# Natural Cellulose from *Ziziphus jujuba* Fibers: Extraction and Characterization

**DOI:** 10.3390/ma16010385

**Published:** 2022-12-31

**Authors:** Aicha Amior, Hamid Satha, Fouad Laoutid, Antoniya Toncheva, Philippe Dubois

**Affiliations:** 1Laboratoire LSPN, Université 8 Mai 1945 Guelma, BP 401, Guelma 24000, Algeria; 2Laboratory of Polymeric and Composite Materials, Materia Nova Materials R&D Center & UMons Innovation Center, 7000 Mons, Belgium

**Keywords:** *Ziziphus jujuba* fibers, cellulose fibers, physicochemical properties, alkaline treatment, thermal stability

## Abstract

Nowadays, due to their natural availability, renewability, biodegradability, nontoxicity, light weight and relatively low cost, natural fibers, especially lignocellulosic fibers, present attractive potential to substitute non-eco-friendly synthetic fibers. In this study, *Ziziphus jujuba* fibers were used, thanks to their low lignin content, as an alternative of renewable resource for the production of cellulosic fibers with suitable characteristics and minimal time and energy consumption. In fact, due to their valuable chemical composition, it was possible to remove the amorphous fractions and impurities from the fiber surface by applying ultrasounds coupled with alkaline treatment (80 °C, 5 wt.% NaOH), followed by a bleaching step. The efficient dissolution of the noncellulosic compounds was confirmed by Fourier Transform Infrared Spectroscopy (FTIR). The resulted increase in the crystallinity index (from 35.7% to 57.5%), occurred without impacting the crystalline structure of the fibers. The morphological analysis of the fibers evidences the higher surface area of the obtained fibers. Based on the obtained results, Ziziphus jujuba fibers were found to present a suitable sustainable source for the production of cellulosic fibers.

## 1. Introduction

The growing environmental concerns and the expected depletion of fossil resources require the design and development of new energy-efficient and environmentally friendly processes and materials. In this spirit, harnessing the abundance and the unique diversity of biomass, effective, renewable and biodegradable raw materials, such as lignocellulosic fibers, present a great interest [1,2]. In addition, these fibers are flexible in processing, lightweight and low-cost compared to conventional synthetic fibers (e.g., carbon, glass, nylon and aramids). As a result of these desirable properties, cellulosic fiber is becoming increasingly popular as a functional biomaterial [3].

Cellulose is the most abundant natural polymer on our planet. Each year, nature produces more than 10^10^ tons of cellulose, representing more than half of the biomass on Earth. In term of chemical structure, cellulose is a linear homopolymer of glucose residues of configuration D, connected in a β-(1–4) glycosidic bond that can be extracted from many sources, such as algae and higher plants; among them, seeds, stalks, leaves, barks, fruits and roots are the most plant parts used [4,5]. Various chemical treatments aiming to isolate the cellulose fibers from raw plants, such as alkali treatment and bleaching, have been explored. These treatments are used to remove the amorphous fractions and impurities from the fiber surface, enhancing its chemical composition and subsequently the final material mechanical properties, surface morphology, crystallinity index and thermal stability [6]. However, due to the considerable time consumption, energy and expense by such treatments, these processes are often combined with some green chemistry extractions based on ultrasound and microwave irradiation approaches [7]. Consequently, the mechanical, optical, thermal and physical properties of the extracted cellulose-based materials are largely determined by the plant source and extraction method [8].

The exceptional physical and chemical properties of cellulose fibers are making them valuable fillers in the concept and development of polymer (nano)composites characterized by high thermomechanical properties [9]. Nonwoven mat fibers, short fibers, woven fibers, microfibrils, nanofibrils and nanocrystals are examples of plant-based fibers known to improve the mechanical properties of polymers. In addition to their excellent mechanical properties (Young’s modulus as high as 138 GPa), cellulosic fibers possess a high aspect ratio and a high active surface, favoring their surface modification for fine-tuned surface chemistry. As a result, their application can be extended to medicine, pharmaceutics, automotives, construction, food packaging and water purification [10,11]. Considering the present environmental and ecological context, it is of common interest to develop new approaches that are oriented towards exploiting natural resources, thus addressing and managing the growing industrial needs, which are not yet aligned with the current cellulosic fiber production.

Interestingly, several reports have dealt with natural cellulosic fiber separation from various natural sources such as artichoke, bamboo, banana, coir, cotton, flax, grass, hemp, jute, okra, pineapple, ramie, sisal and wheat [12]. Even though *Ziziphus jujuba* plants are abundant in different geographic regions, e.g., in China, Iran, Africa, South Korea, Cyprus, Spain, Greece and Sicily, little is known about their natural cellulosic fiber. The *Ziziphus jujuba* plant, also known as jujube, belongs to the Rhamnaceae family, including up to 170 species of *Ziziphus*. In Arab countries, the jujube tree is commonly called sidr, nabk and anneb, while in China it is called Chinese date. Over the recent years, especially in traditional medicine, several parts of *Ziziphus jujuba* have been used in clinical practice as antiurinary agents, antidiabetics, antidiarrhea agents, insomnia agents and sedatives, as well as for infections, bronchitis and hypoglycemic activities [13,14].

The purpose of this study is to evaluate the potential of *Ziziphus jujuba* fiber to replace commonly used non-eco-friendly synthetic fibers across various fields of application. For this purpose and for the first time, the extraction and characterization of a novel variety of cellulose from bleached and delignified *Ziziphus jujuba* fiber as a natural source and as a filler for the development of biodegradable composite materials is proposed. As part of the study, multiple analytical methods (FTIR, TGA, XRD, UV-vis and SEM) were used to investigate the impact of mechanical and chemical treatments on fiber morphology, structure, and thermal characteristics. The obtained fiber properties were also compared to those of other lignocellulosic fibers.

## 2. Materials and Methods

### 2.1. Materials

*Ziziphus jujuba* stems (Figure 1a) were kindly provided by the Faculty of Pharmacy, University Badji Mokhtar Annaba, Algeria. Before their use, they were cut into small pieces, then washed with distilled water to reduce impurities and dried at room temperature. A domestic coffee grinder (Moulinex AR 110830) was used to grind the dry matter to a fine powder (Figure 1b).

Various chemicals with a high degree of purity (>99%), supplied by Aldrich, were used in this work without any further purification: sodium hydroxide, hydrochloric acid, chloroform, methanol, acetone, sodium chlorite and glacial acetic acid. An ultrasound water bath was used for the present experiments.

### 2.2. Isolation of Cellulose

Cellulose was isolated in several steps using the procedure described elsewhere [15,16,17,18], with some specific modifications. Briefly, 20 g of *Ziziphus jujuba* stem powder was preliminary dispersed in a chloroform/methanol (2:1 *v*/*v*) mixture for 2 h to remove all substances soluble in organic solvents (pigments, lipids and waxes) and washed with hot water for 1 h. Next, the obtained powder was submitted to an alkali treatment at 80 °C with NaOH solution (5%) for removing hemicelluloses and lignins. After 8 h, the fibers were rinsed several times with distilled water until reaching neutral pH. To ensure complete delignification, the alkali-pretreated sample was bleached using a solution of sodium chlorite (1.7 wt.% NaClO_2_ in water) and sodium acetate buffer at pH = 4.8 in the ratio 1:1 (*v*/*v*) at 80 °C for 3 h. The operation was repeated until the cellulose fibers turned white and finally lyophilized (−85 °C, 0.0014 mbar, 12 h). All the previously described treatments were performed in an ultrasonic water bath (Elma 100, 40 kHz).

### 2.3. Characterization Methods

#### 2.3.1. Chemical Analysis

The standard NREL technique (NREL/TP-510-42,618), described previously by Sluiter [19], was used for determining the weight content of cellulose, hemicellulose and lignin in the fibers, before and after alkali treatment through high-performance liquid chromatography (HPLC) analysis. Water:acetonitrile was used as mobile phase at 40 °C. The gas pressure was fixed at 3 bar. In brief, the fibers were hydrolyzed with sulfuric acid and the insoluble fraction was used for determining lignin content, while the glucose, xylose and arabinose contents were determined on the soluble fraction. In relation to the initial solid fraction, the weight content of glucose corresponded to the content of cellulose, while the sum of the xylose and arabinose contents was related to the hemicellulose content. 

#### 2.3.2. Fourier Transform Infrared (FTIR) Spectroscopy

FTIR spectra were recorded using a Thermo Scientific Nicolet iS50 spectrometer based on smart iTX-Diamond fitted with an ATR (attenuated total reflectance) system in the range of 500–4000 cm^−1^ with a resolution of 4 cm^−1^ and an accumulation of 32 scans. The samples were placed directly into their compartments, without any prior treatment. The analysis was performed at room temperature.

#### 2.3.3. X-ray Diffraction Analysis (XRD)

XRD analysis was performed under ambient conditions using a Bruker D8 ECO diffractometer with a voltage of 40 kV and an intensity of 25 mA in the range of 5° < 2θ < 80°. The crystallinity index (CI) was obtained according to the empirical Segal method [20], by calculating the ratio of the crystallized area to the total area, as shown in Equation (1):(1)$$\mathrm{CI}\;\%=\left[\frac{I_{200}-{\;I}_{\mathrm{AM}}}{I_{200}}\right]\times 100$$
where I_002_ is the maximum intensity of crystalline region and I_am_ is the lowest intensity of amorphous region. 

The Scherrer Equation (2) was used to calculate crystallite size (CS) [21]:(2)$$\mathrm{CS}=\frac{k\lambda }{\beta \cos\theta }$$
where k is the Scherrer constant (0.9), λ is the X-ray wavelength (0.154 nm), θ is the Bragg angle and β is the peak full width at half maximum (FWHM).

#### 2.3.4. Scanning Electron Microscopy (SEM)

The evolution of fiber morphology (fibrillation and surface morphology) was performed by scanning electron microscopy. The surface morphology of the samples was recorded using a JEOL 840 A LGS with a voltage of 5 kV.

#### 2.3.5. Thermogravimetric Analysis (TGA)

Thermal decompositions of the different fibers were studied by thermogravimetric analysis (TGA) under nitrogen. Approximately 10 mg of the sample was submitted to a temperature ramp from 40 to 800 °C at a heating rate of 10 °C/min using Netzsch STA449 F3 Jupiter, Germany. 

## 3. Results

### 3.1. Chemical Composition

Cellulose, hemicellulose and lignin represent the majority of the lignocellulosic biomass, and their amounts are greatly dependent on the climatic conditions, age of the plant, soil condition and extraction methods, which thus induce a difference in the fibers’ technical performances, such as thermal, mechanical and biodegradable properties [22].

The chemical composition of untreated and alkali treated *Ziziphus jujuba* fibers is compared with various natural fibers in Table 1. The amounts of cellulose, hemicellulose and lignin of the raw fibers were determined to be around 43.0 wt.%, 10.2 wt.% and 5.1 wt.%, respectively. By means of hydrogen bonds and other linkages, cellulose provides high tensile properties to *Ziziphus jujuba* fibers, while hemicellulose and lignin help to maintain the strength of the fiber and to hold water inside for protection against bioattacks.

Alkali treatment changes the chemical composition of fibers significantly. Indeed, the cellulose fraction of *Ziziphus jujuba* fibers increased from 43% to 52% while hemicellulose and lignin fractions were reduced by 4.5% and 2.9%, respectively. This change is due to the breakdown of the ester bonds and α-ether linkages between hydroxyl groups of lignin and carboxylic groups of hemicellulose, which promotes the dissolution of hemicellulose and lignin [23]. Increasing the cellulose content is a desirable effect as cellulose improves the mechanical properties of fibers. In addition, more hemicellulose and lignin can be removed by extending treatment times [24]. However, increasing the concentration of the chemical agents and treatment duration results in lower yields of cellulose fibers due to the breaking of glycosidic bonds, and thus the depolymerization of polysaccharides [25].

Fibers from leaves (abaca) or bast plants (jute, hemp and ramie), which are frequently used in the manufacture of composite materials [26,27,28], generally have the highest cellulose content. These robust fibers clearly possess superior mechanical properties with regards to other plant fibers.

**Table 1 materials-16-00385-t001:** Comparison of chemical composition of *Ziziphus jujuba* fiber before and after alkali treatment with other natural fibers.

Fiber Name	Cellulose (wt.%)	Hemicellulose (wt.%)	Lignin (wt.%)	Reference
Zizyphus jujube	43.0	10.2	5.1	Present study
5% alkali-treated Zizyphus jujube	52.0	5.7	2.2	
Leucas Aspera	50.7	13.2	9.7	[29]
Catharanthus roseus	47.3	9.1	15.1	[30]
Eleusine indica grass	61.3	14.7	11.1	[31]
Abaca	56.0–63.0	15–17	7–10	[26]
Jute	72.0	13.0	13.0	[27]
Hemp	74.0	18.0	4.0	[27]
Ramie	68.6–72.6	13.1–16.7	0.6–0.7	[26]
Saharan aloe vera	67.4	8.2	13.7	[32]
Cotton	85–90	1–3	0.7–1.6	[33]
Sisal	78.0	19.0	8.0	[34]
Bamboo	26–43	30	21–31	[35]
Shwetark	69.6	0.2	16.8	[36]
Flax	81.0	14.0	3.0	[27]
Aerial roots of banyan tree	67.3	13.5	15.6	[37]
Manau rattan (Calamus manan)	42.0	20.0	27.0	[3]
Dracaena reflexa	70.3	11.0	11.4	[38]
Ficus religiosa tree	55.6	13.9	10.1	[39]
Napier grass	45.7	33.7	20.6	[40]
Bagasse	55.2	16.8	25.3	[41]
Cabuya	68–77	4–8	13.0	[42]
Manicaria saccifera palm	74.1	12.0	31.1	[43]
Linum usitatissimum	85.0	9.0	4.0	[44]

### 3.2. Physical Appearance

The evolution of the macroscopic aspect of the fibers at each stage of cellulose isolation is presented in Figure 1. As can be seen, there is a change in the color of the materials obtained after each treatment. These results indicate that the compounds targeted by the applied process have been removed. The effectiveness of the alkaline treatment was indicated by the light brown color, revealing that majority of the hemicelluloses were taken off (Figure 1c). After the bleaching treatment (Figure 1d), the light brown color evolved into a white color, a clear indication of near-pure cellulosic material. In this meaning, it can be concluded that the noncellulose constituents such as lignin, hemicellulose, pectin and wax were effectively removed during the bleaching treatment.

### 3.3. FTIR Analysis 

Analysis by infrared spectroscopy was used to explore the alterations in the functional groups caused by different treatments undergone by *Ziziphus jujuba* fibers. The FTIR spectra of the untreated, the alkali-treated and the cellulose fibers are presented in Figure 2 and the main characteristic bands are summarized in Table 2. 

For all the samples, characteristic O-H stretching vibration bands (3424 cm^−1^) of the inter and intramolecular hydrogen bonds revealing the hydrophilic trend of the fibers were present. In the cellulose fiber, this characteristic band was more intense, indicating higher cellulose content due to the elimination of lignin and hemicellulose [45]. The characteristic band at 2922 cm^−1^ was attributed to the stretching vibration of the sp^3^ C-H group, and the H-O-H stretching vibration as result of the absorbed water appeared at 1640 cm^−1^. A more pronounced intensity was found in the case of cellulose as compared to both untreated and alkali-treated fibers due to the higher moisture content in this sample [46]. Furthermore, the extraction of the amorphous phase (lignin and hemicellulose) after chemical treatments was confirmed by the disappearance of the compound characteristic absorption bands: (i) the first one at 1734 cm^−1^ corresponds to the C=O stretching vibration of p-coumaric acids of lignin and/or hemicellulose; (ii) a second band related to either acetyl or ester linkages of carboxylic groups of the ferulic or uronic ester groups in hemicellulose [47]; (iii) the other absorption bands at 1508 cm^−1^ and 1236 cm^−1^ correspond to the C=C bond strain of the aromatic ring of lignin and the C-O stretching of the aryl group C=O-O vibration of acetyl groups, respectively. The disappearance of these bands is due to the breakage of carboxylic ester bonds and the oxidation of terminal glucopyranose units [47]. The two bands at 1429 cm^−1^ and 1327 cm^−1^ are attributed to the vibrations of C-C bonds in the CH_2-_, CH_3_ groups and to the C-O skeletal vibrations, respectively, while the band at 1054 cm^−1^ is assigned to the ether C-O binding vibrations [48]. The C-O-C pyranose ring skeletal vibration appears at 1031 cm^−1^. The increase in the percentage of cellulosic components after the successive pretreatments is evidenced by the presence of the typical peaks ascribed to cellulose at 1159 cm^−1^ and 897 cm^−1^, corresponding to the asymmetric stretching vibration of the C-O-C and the β-(1→4)-glycosidic bonds of glucose rings [49,50]. 

### 3.4. Crystallinity Analysis

Based on previously published data, it is known that the quantity of cellulose component of fibers affects their crystallinity [51]. Accordingly, it was of importance to study this parameter in detail, and X-ray diffraction analysis was performed. The X-ray diffractograms of untreated, alkali-treated and cellulose fibers are shown in Figure 3. The degree of crystallinity varies considerably with the fiber origin and with the physical and chemical treatments to which it has been subjected [51].

The general profile of the three diffractograms is similar. The 2θ values ranging between 10° and 40° display same main peaks at 15.0°, 16.20°, 21.85° and 35.4°. These 2θ values were ascribed to the following crystallographic planes: (1–10), (110), (002) and (004), corresponding to the typical structure of cellulose I, conferring the material rigid nature. The existence of other fractions such as lignin, hemicellulose, amorphous cellulose and pectin in the fibers was confirmed by the presence of diffraction peaks at 2θ values of 16.2°, while the content of α-cellulose was assigned by the peak at 2θ values of 21.85° [52]. In general, the direction and alignment of the fibers are signed by the peak 2θ located at 35.4° [53]. Since they contain diffraction peaks characteristic for the amorphous phase (16.2°), as well as diffraction peaks unique for the crystalline phase (15°, 21.85° and 35.4°), it can be concluded that the obtained diffractograms are typical of semicrystalline material. The diffraction peaks at 15° and 16.2° were assigned to the principal equatorial planes indexed at (1–10) and (110) in the monoclinic cell with two chains. For a high percentage of cellulose I (high crystallinity degree), these two diffraction peaks were present and distinct from each other. Conversely, when the fibers contain a high percentage of amorphous phase (lignin, pectin and hemicellulose fraction), a single common diffraction peak is observed. 

The structural transformation from cellulose I to cellulose II was not observed in the present study due to the low alkaline concentration used (5 wt.%). This result agreed with the finding described by Yue et al. [54], where the conversion from cellulose I to cellulose II could not be reached when treating cellulose-based fibers with less than 10 wt.% of NaOH. Based on the obtained findings, it can be concluded that the structure of cellulose I was largely preserved in all the samples. The crystallinity indices are determined for the various samples and reported in Table 3. Crystallinity index is a factor used to exemplify the relative quantity and the order of the crystallites in the fibers [51].

According to Table 3, the untreated, alkali-treated and cellulose fibers exhibit a crystalline index of 35.70%, 50.81% and 57.51%, respectively. As a result of the successive elimination of low-molecular-weight compounds (amorphous lignin, hemicellulose and impurities—waxes and pectin) by the applied alkali and bleaching treatments, the amount of crystalline cellulose fraction proved to be enhanced. As a result, the crystals became more regularly arranged and favored an increase in the CI values, as shown by the higher (200) reflection intensity in chemically treated fibers with respect to the raw fibers.

This finding was in good agreement with the results reported by Zhao et al. [23] and Ciftci et al. [55], for cellulose fibers from poplar wood and canola straw, respectively. It is expected that the strength and stiffness of the fibers are influenced by the treatment steps and they are increased by the improved crystallinity [56].

Crystal size is also known to affect water absorption capacity of the fiber [51]. The average size of a sole crystal was found to be 16.3 nm, 12.81 nm and 10.12 nm for untreated, alkali-treated and cellulose fibers, respectively. The larger CS of raw fiber may be due to the presence of aggregate precipitation and some larger filaments, which affect the chemical reactivity and the mechanical strength of the fibers. A reduction in the crystallite sizes after the surface treatment was observed. In addition, the decrease in the crystallite sizes affected the constricted packing of crystallites and considerably decreased the moisture permeation, which reduced the hydrophilic behavior of the fiber. The crystallite sizes of the fiber may be increased or decreased depending on the source of the fiber and the extraction process, as reported by Cullity [57]. The high crystallinity index and low crystal size of the cellulosic fiber were found to be favorable factors for producing durable biocomposites [58].

### 3.5. Thermal Properties

The thermal stability of natural fibers is an important property that determines their suitability for different industrial applications and compatibility with polymer melt processing conditions. In order to evaluate whether the chemical treatment induced changes in the thermal stability of the fibers, a TGA was performed on the untreated, alkali-treated and cellulose fibers, as shown in Figure 4. The onset (T_onset_) and the maximum (T_max_) degradation temperatures of the samples are listed in Table 4. 

The difference in the thermal stability of lignocellulosic fibers can be attributed to some variation of the fiber chemical structure and related physical properties [58]. One can observe that the three fibers present some difference in weight loss below 100 °C, i.e., 6.8 wt.%, 3.9 wt.% and 3.1 wt.% for untreated, alkali-treated and cellulose fibers, respectively. This weight loss corresponds to the release of both moisture and bound water [59]. The thermal degradation of the other constituents occurs at different temperature ranges. Cellulose is more thermally resistant than hemicellulose because of the decarboxylation of the glycosidic linkages [60]; its degradation occurs between 300 °C and 420 °C, while hemicellulose degrades between 190 °C and 300 °C [60]. Lignin thermal degradation is a complex process that takes place over a wide temperature range between 200 °C and 600 °C, occuring in different steps due to its intricate structure with numerous branches of oxygen-based functional groups such as phenolics and aromatics [61]. Untreated fibers present thermal stability lower than the other two fibers due to the presence of hemicellulose in the fibers before treatment.

Chemically-treated fibers have one degradation stage, while untreated fibers visibly display two stages associated to the degradation of numerous components such as cellulose, hemicellulose and lignin, which are usually renowned for their dissimilar degradation temperatures. The first stage appears in the DTG thermograms as a shoulder between 175 °C and 275 °C and the second one appears as a prominent peak between 275 °C and 375 °C. Therefore, the chemically treated fibers revealed higher onset decomposition at 215 °C and 220 °C for the alkali-treated and cellulose fibers, respectively, when compared to the raw fibers at 175 °C. This increase in the onset thermal degradation temperature is ascribed to the elimination of amorphous hemicellulose and lignin, known for their low decomposition temperatures. It is worth noting that the onset degradation temperature of cellulose fibers is influenced by the crystallinity index, degree of polymerization and hydrogen bond energy [62]. The thermal stability of the untreated Ziziphus jujuba fiber is relatively lower than the chemically treated fibers. 

### 3.6. Morphological Analysis 

As a next step, it was important to examine the surface morphology of Ziziphus jujuba fiber before and after different treatments. This approach will allow one to understand the distribution of various constituents present over the fiber surface. The SEM analysis was performed in the longitudinal direction at different magnifications, as shown in Figure 5. 

As evidenced from SEM micrographs, the alkali and bleach treatments of the fibers induced some morphological changes as a result of the dissimilarity in the operational conditions.

Before any treatment, the cellulose chains are oriented and joined together by cement components (lignin and hemicellulose), forming thickly packed fiber bundles of a certain diameter with a complex structure. Significant magnification of 1 mm (Figure 5a,b) shows that the Ziziphus jujuba fiber is quite cylindrical, which grows the specific area and thus is nepotistic to the chemical processes. In addition, the raw fiber (Figure 5a) showed a partially smooth surface due to the deposition of a waxy layer on its outside [63]. Some discrete white-colored and undersized patches adhered on the surface of the fibers, which may be impurities or noncellulosic elements. The mechanical properties, and thus the interfacial union for biocomposites, are conditioned by the presence of these impurities. With partial elimination of some quantities of outer-layer components such as hemicellulose, lignin, wax and other impurities, many irregularities are present on the surface of the alkali-treated fibers, increasing their roughness texture compared to the raw fibers (Figure 5b). The increase in roughness is favorable for composite materials due to the enhanced adherence to the fiber with the matrix. An enhancement of flax/epoxy polymer interfacial adhesion by alkali treatment was reported by Yan et al. [64]. After NaOH treatment, a slight variation in the diameter of fiber bundles was observed as result of partial removal of lignin. Lignin is a component displaying binding ability in fibers and constituting a bridge link with cellulose, thus preserving its bundle-like morphology after NaOH treatment [65]. As presented in Figure 5c,d the bundles of cellulose are well divided into individual microsized fibers via consecutive elimination of lignin (binding material) and thus breaking the bond between them by bleaching treatment. These individual fibers exhibit an unaligned and random structure. They are entangled and superimposed, with uneven size. The length of the observed fibers (a few hundred µm) always remains much greater than their width (from 25 to 30 µm). The reactivity and the good mechanical adhesion with other polymeric materials could be favored by the presence of different sizes of cellulose fibrils [64]. 

## 4. Conclusions

In the present work, cellulose fibers were successfully extracted from a natural source, *Ziziphus jujuba*, using ultrasounds as an alternative energy source for alkali and bleaching treatments. Increasing cellulose while decreasing the hemicellulose and lignin content in the fibers validated the efficiency of the investigated chemical treatment. This was confirmed by the FTIR, where the disappearance of the characteristic bands of lignin and hemicellulose (characteristic band located at 1734 cm^−1^ attributed to the C=O stretching vibration of p-coumaric acids of lignin and/or hemicellulose). The characteristic bands were located at 1508 cm^−1^ (C=C of aromatic rings of lignin) and at 1236 cm^−1^ (C-O groups of methoxy of lignin). In addition, the purified cellulose fibers preserved the structure of cellulose I with nanometric size (10.12 nm), higher crystallinity (57.50%) and good thermal stability (220 °C). Based on the morphological analysis, it can be concluded that the *Ziziphus jujuba* fibers were well divided into individual microsized fibers with random structures and have the advantage of retaining a crystalline (useful for strength) and amorphous structure (interesting for chemical reactivity). The present results reveal the great potential of such cellulose-based fibers as fillers for the development of a new generation of sustainable biocomposites with biodegradable character. Cellulose fibers from *Ziziphus jujuba* could be also modified through grafting phosphorous compounds, and serve for the development of flame-retardant biobased composites.

## Figures and Tables

**Figure 1 materials-16-00385-f001:**
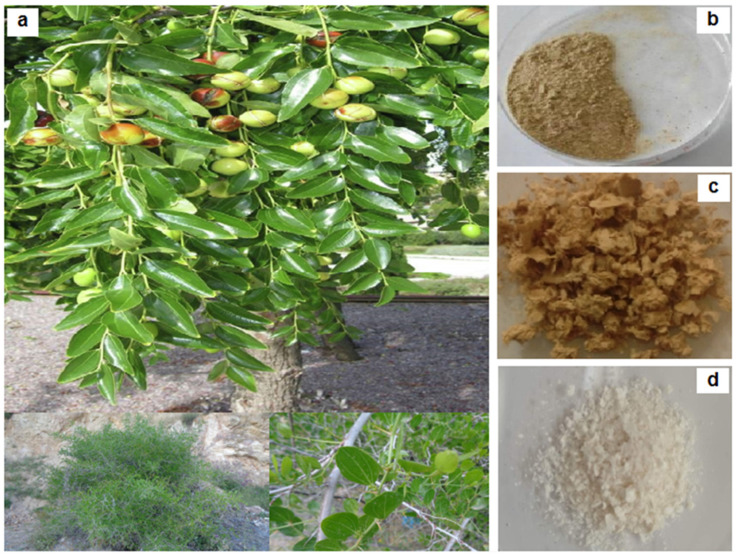
Photographs of (**a**) *Ziziphus jujuba* plant, (**b**) untreated *Ziziphus jujuba* material, (**c**) alkali-treated samples and (**d**) final bleached cellulose.

**Figure 2 materials-16-00385-f002:**
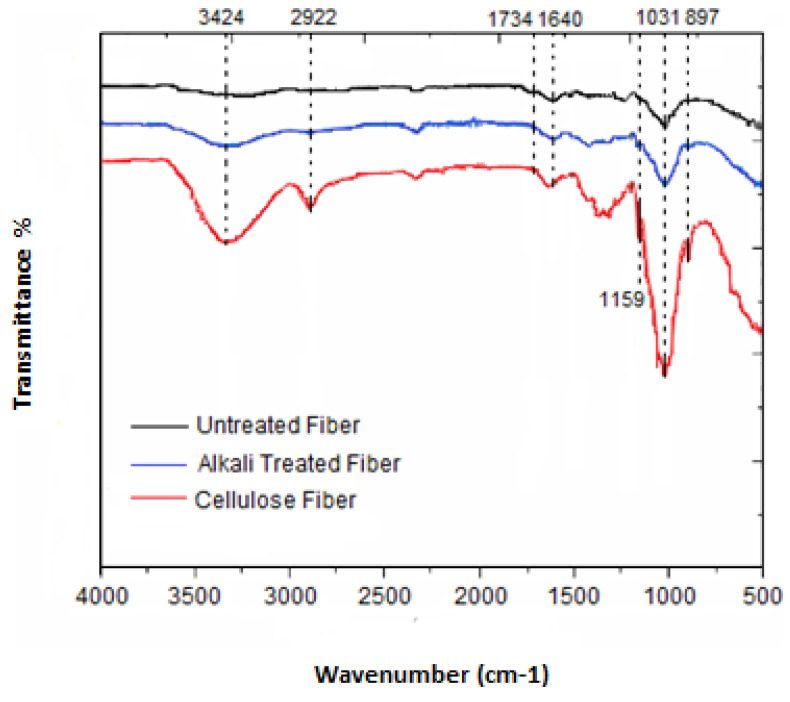
FTIR spectra of untreated, alkali-treated and cellulose fibers.

**Figure 3 materials-16-00385-f003:**
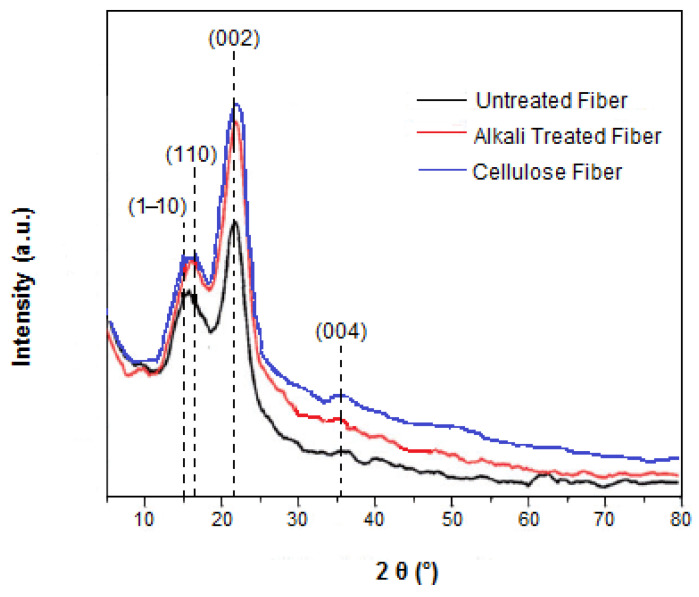
X-ray diffractograms of untreated, alkali-treated and cellulose fibers.

**Figure 4 materials-16-00385-f004:**
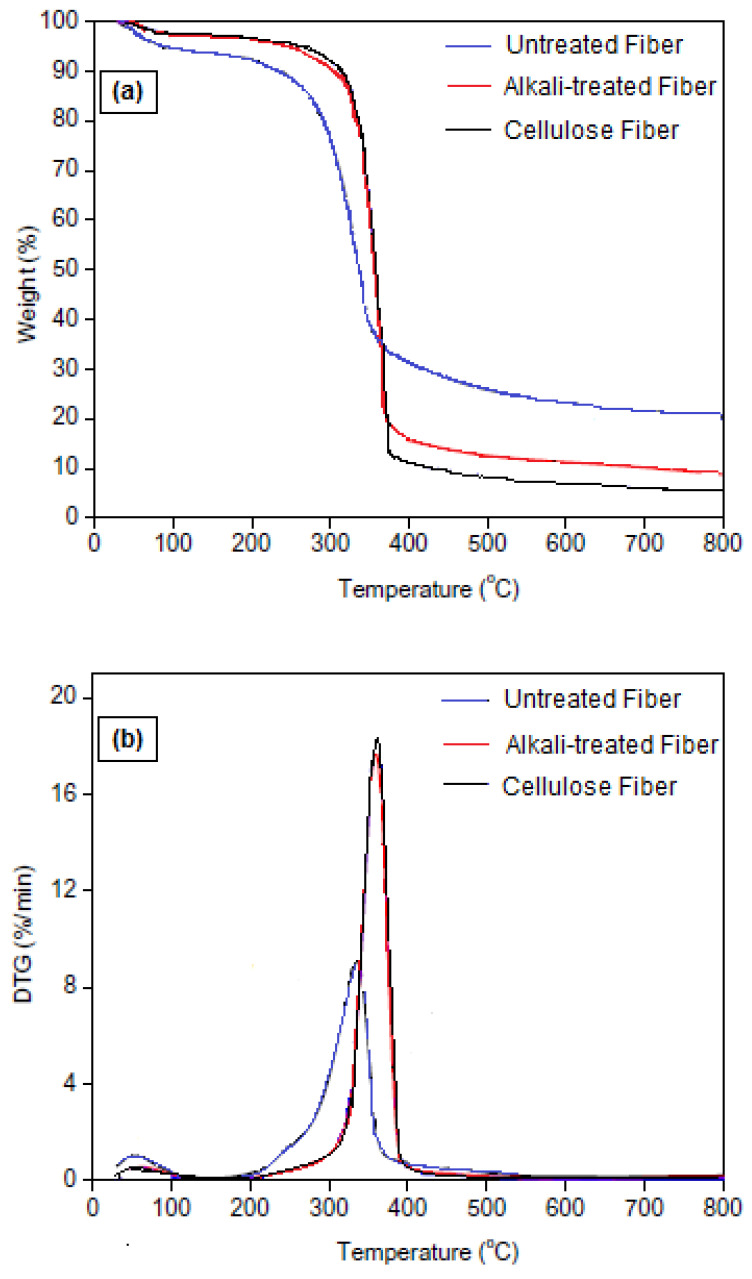
(**a**) TG and (**b**) DTG thermograms of the untreated, alkali-treated and cellulose fibers.

**Figure 5 materials-16-00385-f005:**
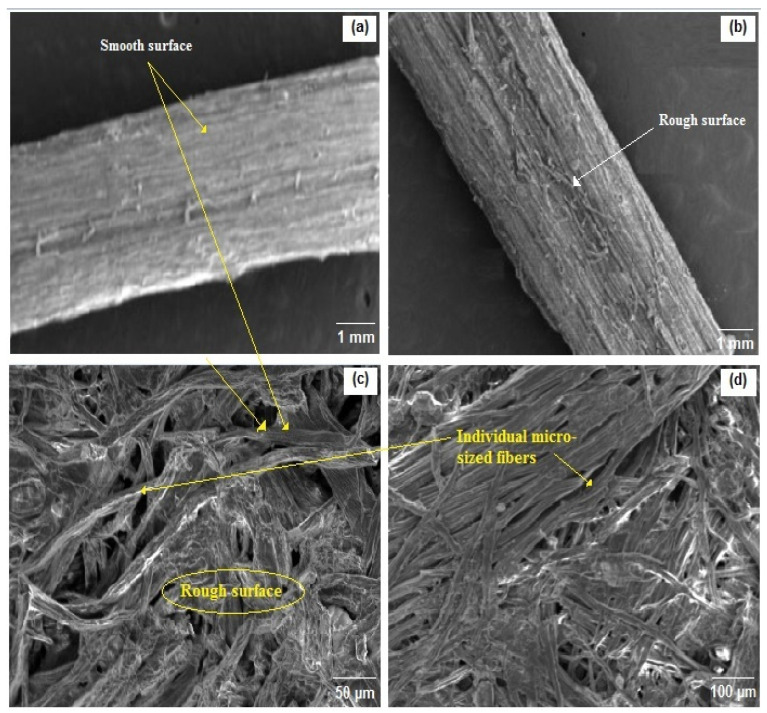
SEM micrographs of (**a**) untreated, (**b**) alkali-treated and (**c**,**d**) cellulose fibers.

**Table 2 materials-16-00385-t002:** Characteristic bonds of different compounds of *Ziziphus jujuba* fiber.

Characteristic Bands (cm^−1^)		Cellulose	Lignin	Hemicellulose
3424 2922 1734 1508 1429 1054	-OH C-H C=O C=C CH_2_, CH_3_ C-O, C-O-C	✓ ✓	✓ ✓	- ✓
		-	✓	✓
		-	✓	-
		✓	✓	✓
		✓	-	-

**Table 3 materials-16-00385-t003:** Crystallinity index and crystallite sizes of untreated, alkali-treated and cellulose fibers.

Samples	Crystallinity Index (%)	Crystallite Size (nm)
Untreated fiber	35.70	16.30
Alkali-treated fiber	50.81	12.81
Cellulose fiber	57.51	10.12

**Table 4 materials-16-00385-t004:** Thermal degradation temperatures and charred residue of *Ziziphus jujuba* fibers before and after different treatments (10 °C/min, under N_2_).

Samples	T_onset_ (°C)	T_max_ (°C)	Charred Residue (%)
Untreated fiber	175	335	19.1
Alkali-treated fiber	215	360	8.1
Cellulose fiber	220	362	5.2

## Data Availability

Not applicable.

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
