# Peer review of "Natural Cellulose from Ziziphus jujuba Fibers: Extraction and Characterization"

_materials, 2022, doi:10.3390/ma16010385_

Round 1

Reviewer 1 Report

The paper describes extraction of cellulose from Ziziphus jujuba plant. The authors attempted chemical methods to extract fibers from this otherwise less explored plant source. The study is good and relevant to the scope of the journal. Few changes are required:

1.     The paper needs to be revised to remove language errors

2.     Introduction section should state about the possible utilization of this fibers in the country of origin and/or at global level.

3.     It will be good if authors can compare the yield analysis from this plant using their methods with appropriate plant sources and similar extraction methods.

4.     L9: Look for spelling error

5.     L9-11: Rephrase the sentence

6.     L18: Correct the verb

7.     Write in abstract why this plant was selected?

8.     Write lignocellulosic. Look for spelling mistake throughout the paper

9.     L36: Do not write nature with a capital letter

10.  L40: Cellulose cannot be extracted from all the organisms listed here. Look for the correct information and do not confuse cellulose with cellulase.

11.  L46: What is ‘sue’?

12.  L60: Correct the verb form

13.  L76-77: Use correct medical/clinical terms

14.  L95-96: Why the names of these chemicals are given without any context?

15.  L113: Do not write composition of buffer this way. Mention its molarity or ionic strength

16.  L121, L133. L182: Use correct form of verb

17.  To represent FTIR spectra use correct form of inverse symbols

18.  L361: Correct the typo

Author Response

Thank you so much for your recommendations and the reviewers’ comments on our manuscript. We have revised the manuscript according to most of the reviewers’ comments (changes were highlighted in red), with detailed response given below.

  • Ethical question asked by the editor.

“Ziziphus jujuba stems Zizyphus were kindly provided by the faculty of pharmacy-University Badji Mokhtar Annaba-Algeria.”

This sentence was added in Materials section

Reviewer1 :

  1. The paper needs to be revised to remove language errors : The paper has been rewritten and the English improved. Hope this new draft will satisfy the reviewers.
  2. Q. Introduction section should state about the possible utilization of this fibers in the country of origin and/or at global level.

R: The following information has been added in the introduction part:

L 69-72: - Over the recent years, especially in traditional medicine, several parts of Ziziphus jujuba have been used in clinical practice for their as anti-urinary agents, anti-diabetics, infec-tions, anti-diarrhea, insomnia agents, sedatives, bronchitis and hypoglycemic activities.

  1. Q: It will be good if authors can compare the yield analysis from this plant using their methods with appropriate plant sources and similar extraction methods.

R: We found some articles where the yield but we could not compare with these values since they were obtained using different experimental methods.

  1. Q: L9: Look for spelling error

R: “lingo-cellulosic” corrected to “ligno-cellulosic”

  1. Q: L9-11: Rephrase the sentence

R: It’s corrected from

The paper has been rewritten and the English improved. Hope this new draft will satisfy the reviewers.

  1. L18: Correct the verb

R: The paper has been rewritten and the English improved.

  1. Write in abstract why this plant was selected?

R: We mention in the abstract that Ziziphus jujuba fibers were selected owing to their low lignin content.

  1. Q: Write lignocellulosic. Look for spelling mistake throughout the paper

R: The paper has been rewritten and the English improved.

  1. Q: L36: Do not write nature with a capital letter

Response: It’s corrected from “Nature” to “nature”.

  1. Q: L40: Cellulose cannot be extracted from all the organisms listed here. Look for the correct information and do not confuse cellulose with cellulase.

Response: Thanks for the correction. We removed bacteria as a source of cellulose.

  1. L46: What is ‘sue’?

Response: L46: It is a mistake, ‘sue to’‘ has been replaced by ‘due to’

  1. L60: Correct the verb form

Response: The paper has been rewritten and the English improved.

  1. L76-77: Use correct medical/clinical terms

R: Thanks for the comment. We used the correct terms in the following sentences.

Over the recent years, especially in traditional medicine, several parts of Ziziphus jujuba have been used in clinical practice for their as anti-urinary agents, anti-diabetics, infections, anti-diarrhea, insomnia agents, sedatives, bronchitis and hypoglycemic activities [13, 14].

  1. L95-96: Why the names of these chemicals are given without any context?

Response: This is the continuation of the previous sentence. “Various chemicals with a high degree of purity (> 99%), supplied by Aldrich, were used in this work”

  1. L113: Do not write composition of buffer this way. Mention its molarity or ionic strength

Response: It’s corrected from;

“with a sodium acetate buffer at pH= 4.8 (27g of NaOH, the addition of 75ml of glacial acetic acid for 1 L of distilled water)”

To

“with a sodium acetate buffer at pH= 4.8 in the ratio 1:1 (v/v)”

  1. L121, L133. L182: Use correct form of verb

Response: The paper has been rewritten and the English improved.

  1. To represent FTIR spectra use correct form of inverse symbols

Response: the following modifications have been done in the document:

3424 cm-1 (3424 cm-1)

2922 cm−1 (2922 cm−1)

sp3 (sp3)

1640 cm-1 (1640 cm-1)

1734 cm-1 (1734 cm-1)

1508 cm-1 (1508 cm-1)

1236 cm-1 (1236 cm-1)

1429 cm-1 (1429 cm-1)

CH2, CH3 (CH2, CH3)

1327 cm-1 (1327 cm-1)

  1. L361: Correct the typo

As a next step it was of importance to examine the surface morphology of Ziziphus fiber before and after different treatments.

Response: The paper has been rewritten and the English improved.

Reviewer 2 Report

In this paper, Ziziphus jujuba fibers were used as an alternative of renewable resource for the production of cellulosic fibers with suitable characteristics. The topic of this paper involves the frontier of the discipline, which has important research significance and a wide range of application prospects. In general, the work is well done, while the conclusion is supported by the experimental and results. However, there are still some issues to be addressed before its acceptance.

1.        It should be noted that the manuscript needs to be carefully edited by someone with expertise in technical editing in English, with special attention to English grammar, spelling, and sentence structure.

2.        When introducing the structure, properties, modifications and applications of cellulose, some recent articles should be included: Source of Nanocellulose and Its Application in Nanocomposite Packaging Material: A Review; Potential new material for optical fiber: Preparation and characterization of transparent fiber based on natural cellulosic fiber and epoxy; etc.

3.        Besides low lignin content, does jujube fiber have other advantages? As can be seen in Table 1, there are fibers with lower lignin content.

4.        The author said: “There is a change in the obtained materials color after the applied treatment, indicating that the compounds targeted by the applied treatment were removed.” The author can add micro analysis of powder samples before and after treatment, so that readers can see the treatment effect more directly.

5.        In the table, more examples of natural fibers should be included for better comparison: Characterization of natural fiber from manau rattan (Calamus manan) as a potential reinforcement for polymer-based composites; Plant extract-loaded bacterial cellulose composite membrane for potential biomedical applications; etc.

6.        References should be labeled for infrared spectral analysis.

7.        Non-ultrasonic water bath can be used for delignification as a comparative experiment to emphasize the novelty of this paper.

Author Response

Thank you so much for your recommendations and the reviewers’ comments on our manuscript. We have revised the manuscript according to most of the reviewers’ comments (changes were highlighted in red), with detailed response given below.

Reviewer2:

  1. It should be noted that the manuscript needs to be carefully edited by someone with expertise in technical editing in English, with special attention to English grammar, spelling, and sentence structure.

Response: the English of the paper has been improved. We hope that this new version will satisfy the reviewers. Thank you for the time spent in the examination of our paper.

  1. When introducing the structure, properties, modifications and applications of cellulose, some recent articles should be included: Source of Nanocellulose and Its Application in Nanocomposite Packaging Material: A Review; Potential new material for optical fiber: Preparation and characterization of transparent fiber based on natural cellulosic fiber and epoxy; etc.

Response: These following articles were included;

- Ding, L.; Han, X.; Cao, L.; Chen, Y.; Ling, Z.; Han, J.; He, S.; Jiang, S. Characterization of natural fiber from manau rattan (Calamus manan) as a potential reinforcement for polymer-based composites. J. Bioresour. Bioprod. 2022, 7, was added in the introduction as reference N°3.

- Wang, J.; Han, X.; Zhang, C.; Liu, K.; Duan, G. Source of Nanocellulose and Its Application in Nanocomposite Packaging Material: A Review. Nanomaterials. 2022, 12, 3158, was added in the introduction as reference N°10.

- Han, X.; Ding, L.; Tian, Z.; Song, Y.; Xiong, R.; Zhang, C.; Han, J.; Jiang, S. Potential new material for optical fiber: Preparation and characterization of transparent fiber based on natural cellulosic fiber and epoxy. Int. J. Biol. Macromol. 2022, was added in the introduction as reference N°11.

  1. Besides low lignin content, does jujube fiber have other advantages? As can be seen in Table 1, there are fibers with lower lignin content.

Response: till know, we are not able to answer this question. We did not identify other advantage of jujube fibers. Hope further results we are conducting on these fibers will enable us to identify them. Contrary to coton and flax that are already used and valorized in textiles and other applications, jujube fibers are not valorized at this time.

  1. The author said: “There is a change in the obtained materials color after the applied treatment, indicating that the compounds targeted by the applied treatment were removed.” The author can add micro analysis of powder samples before and after treatment, so that readers can see the treatment effect more directly.

Response: Yes, we agree. Images presented in the analysis section present the Morphological analysis of these fibers.

Figure 5. SEM micrographs of (a) untreated, (b) alkali treated and (c-d) cellulose fibers.

  1. In the table, more examples of natural fibers should be included for better comparison: Characterization of natural fiber from manau rattan (Calamus manan) as a potential reinforcement for polymer-based composites; Plant extract-loaded bacterial cellulose composite membrane for potential biomedical applications; etc.

Response: We added the article below (reference N°3) in the introduction section and in the table1 for further information and better comparison.

Ding, L.; Han, X.; Cao, L.; Chen, Y.; Ling, Z.; Han, J.; He, S.; Jiang, S. Characterization of natural fiber from manau rattan (Calamus manan) as a potential reinforcement for polymer-based composites. J. Bioresour. Bioprod. 2022, 7.

  1. References should be labeled for infrared spectral analysis.

Response: References were labeled for infrared spectral analysis as follow:

For all the samples, characteristic O-H stretching vibrations band (3424 cm-1) of the inter and intra-molecular hydrogen bonds revealing the hydrophilic trend of the fibers are present. In the cellulose fiber, this characteristic band was more intense indicating higher cellulose content due to the elimination of lignin and hemicellulose [45]. The characteristic band at 2922 cm−1 was attributed to the stretching vibration of sp3 C-H group and the H–O–H stretching vibration as result of the absorbed water appeared at 1640 cm-1. A more pronounced intensity was found in the case of cellulose as compared to both untreated and alkali-treated fibers due to the higher moisture content in this sample [46]. Furthermore, the extraction of the amorphous phase (lignin and hemicel-lulose) after chemical treatments was confirmed by the disappearance of the compound characteristic absorption bands: i) the first one at 1734 cm-1 corresponding to the C=O stretching vibration of p-coumaric acids of lignin and/or hemicellulose, ii) a second band related to either acetyl or ester linkages of carboxylic groups of the ferulic or uronic ester groups in hemicellulose [47] while iii) the other absorption bands at 1508 cm-1 and 1236 cm-1 correspond to the C=C bond strain of the aromatic ring of lignin and the C–O stretching of the aryl group C=O-O vibration of acetyl groups, respectively. The disap-pearance of these bands is due to the breakage of carboxylic ester bonds and the oxida-tion of terminal glucopyranose units [47]. The two bands at 1429 cm-1 and 1327 cm-1 are attributed to the vibrations of C–C bonds in CH2-, CH3 groups and to the C–O skeletal vibrations, respectively, while the band at 1054 cm-1 is assigned to the ether C-O binding vibrations [48]. The C–O–C pyranose ring skeletal vibration appears at 1031 cm-1. The increase in the percentage of cellulosic components after the successive pretreatments is evidenced by the presence of the typical peaks ascribed to cellulose at 1159 cm-1 and 897 cm−1, corresponding to the asymmetric stretching vibration of the C–O–C and the β-(1→4)-glycosidic bonds of glucose rings [49, 50].

  1. Non-ultrasonic water bath can be used for delignification as a comparative experiment to emphasize the novelty of this paper.

Response: Unfortunately, we did not perform this study but we agree that this kind of comparison could present some interest, maybe in coming study.

Reviewer 3 Report

1. Please do a language editing.

2.  Please provide more details for the extraction of cellulose. 

3. For the coffee grinder mentioned in section 2.1 please provide the model/make.

4. What is the inference from Table 1?

5. Add the limitations, advantages, and the future scope of the work in the conclusions section. 

Author Response

Thank you so much for your recommendations and the reviewers’ comments on our manuscript. We have revised the manuscript according to most of the reviewers’ comments (changes were highlighted in red), with detailed response given below.

Reviewer 3:

  1. Please do a language editing.

R: the English of the paper has been improved.

  1. Please provide more details for the extraction of cellulose. 

R: The following section has been enhanced:

Cellulose was isolated in several steps using the procedure described elsewhere [15, 16, 17, 18], with some specific modifications. Briefly, 20g of Ziziphus jujuba stem powder was preliminary dispersed in a chloroform/methanol (2:1 v/v) mixture for 2h, for removing all substances soluble in organic solvents (pigments, lipids and waxes) and washed with hot water for 1h. Next, the obtained powder was submitted to an alkali treatment at 80°C with NaOH solution (5%) for removing hemicelluloses and lignins. After 8h, the fibers were rinsed several times with distilled water until reaching neutral pH. To ensure complete delignification, the alkali-pretreated sample was bleached us-ing a solution of sodium chlorite (1.7 wt% NaClO2 in water) and sodium acetate buffer at pH= 4.8 in the ratio 1:1 (v/v) at 80°C for 3 h. The operation was repeated until the cel-lulose fibers turned white and finally lyophilized (−85 °C, 0.0014 mbar, 12h). All the previously described treatments were performed in an ultrasonic water bath (Elma 100, 40 kHz).

  1. For the coffee grinder mentioned in section 2.1 please provide the model/make.

Response: we used Moulinex AR 110830 grinder. This information has been added in the manuscript

  1. What is the inference from Table 1?

R: Table 1 presents the chemical composition of untreated and alkali treated Ziziphus jujuba fibers in comparison with various natural fibers. This table has been completed as recommended by reviewer 1 for providing a comparison with the composition with other well-known fibers.

  1. Add the limitations, advantages, and the future scope of the work in the conclusions section. 

Response: we completed the conclusion part by adding the following information:

The present results are revealing the great potential of such cellulose-based fibers as fillers for the development and processing of new generation of sustainable biocompo-sites with biodegradable character. Cellulose fibers from Ziziphus jujuba could be also modified through grafting phosphorous compound and serve for the development of flame-retardant biobased composites.

Reviewer 4 Report

Figure 1 is of low quaility and needs to be in higer resolution

there are numerus errors related to the writing of CH3, where 3 needs to be written as subsvript and cm-1 where -1 needs to be written in superscript.

The FTIR results ahould be accompined by the proper refrences, not only in the end of the paragraph

How come the FTIR did not detect the crystalline/amporphus regions of the cellulose fibers? Why can't you connect both the FTIR and CI results together?

Author Response

Thank you so much for your recommendations and the reviewers’ comments on our manuscript. We have revised the manuscript according to most of the reviewers’ comments (changes were highlighted in red), with detailed response given below.

Reviewer 4:

Figure 1 is of low quaility and needs to be in higher resolution

Response: the figure 1 was replaced using new images with high resolution.

there are numerus errors related to the writing of CH3, where 3 needs to be written as subsvript and cm-1 where -1 needs to be written in superscript.

R: the paper has been carefully examined and the English improved.

The FTIR results ahould be accompined by the proper refrences, not only in the end of the paragraph;

R: References were labeled for infrared spectral analysis as follow:

For all the samples, characteristic O-H stretching vibrations band (3424 cm-1) of the inter and intra-molecular hydrogen bonds revealing the hydrophilic trend of the fibers are present. In the cellulose fiber, this characteristic band was more intense indicating higher cellulose content due to the elimination of lignin and hemicellulose [45]. The characteristic band at 2922 cm−1 was attributed to the stretching vibration of sp3 C-H group and the H–O–H stretching vibration as result of the absorbed water appeared at 1640 cm-1. A more pronounced intensity was found in the case of cellulose as compared to both untreated and alkali-treated fibers due to the higher moisture content in this sample [46]. Furthermore, the extraction of the amorphous phase (lignin and hemicel-lulose) after chemical treatments was confirmed by the disappearance of the compound characteristic absorption bands: i) the first one at 1734 cm-1 corresponding to the C=O stretching vibration of p-coumaric acids of lignin and/or hemicellulose, ii) a second band related to either acetyl or ester linkages of carboxylic groups of the ferulic or uronic ester groups in hemicellulose [47] while iii) the other absorption bands at 1508 cm-1 and 1236 cm-1 correspond to the C=C bond strain of the aromatic ring of lignin and the C–O stretching of the aryl group C=O-O vibration of acetyl groups, respectively. The disap-pearance of these bands is due to the breakage of carboxylic ester bonds and the oxida-tion of terminal glucopyranose units [47]. The two bands at 1429 cm-1 and 1327 cm-1 are attributed to the vibrations of C–C bonds in CH2-, CH3 groups and to the C–O skeletal vibrations, respectively, while the band at 1054 cm-1 is assigned to the ether C-O binding vibrations [48]. The C–O–C pyranose ring skeletal vibration appears at 1031 cm-1. The increase in the percentage of cellulosic components after the successive pretreatments is evidenced by the presence of the typical peaks ascribed to cellulose at 1159 cm-1 and 897 cm−1, corresponding to the asymmetric stretching vibration of the C–O–C and the β-(1→4)-glycosidic bonds of glucose rings [49, 50].

How come the FTIR did not detect the crystalline/amorphus regions of the cellulose fibers? Why can't you connect both the FTIR and CI results together?

Response: we agree with reviewer that some comparison and correlation could be done between FTIR and XRD results obtained on cellulose.

Round 2

Reviewer 4 Report

All changes have been made.